# Addressing Water Stress and Climate Variability in the Mediterranean: Study of Regulated Deficit Irrigation (RDI) and Non-Irrigation (NI) in Tempranillo and Cabernet Sauvignon (*Vitis vinifera* L.)

Antoni Sánchez-Ortiz, Miriam Lampreave * and Maria Assumpta Mateos

Departament de Bioquímica i Biotecnologia, Facultat d'Enologia de Tarragona, Universitat Rovira i Virgili, Campus Sescelades, Marcellí Domingo s/n, 43007 Tarragona, Spain; antonio.sanchezo@urv.cat (A.S.-O.); mariaassumpta.mateos@urv.cat (M.A.M.)
* Correspondence: miriam.lampreave@urv.cat; Tel.: +34-629611532

**Abstract:** Climate variability in Mediterranean viticultural areas, primarily attributed to climate change, will significantly impact water requirements, consequently leading to changes in irrigation management. The primary aim of this study was to assess the response of the Tempranillo and Cabernet Sauvignon grape varieties when subjected to deficit drip irrigation (RDI), in comparison to non-irrigation (NI), during various climatic years. The defined irrigation strategies involved water application equivalent to 35% and 80% of the (ET0 (reference crop evapotranspiration) × Kc (crop coefficient)). The ecophysiology of both grapevines was evaluated through the measurement of stomatal conductance (gs), sap flow, transpiration, leaf water potential (LWP), and $CO_2$ assimilation (A). Additionally, essential parameters including the crop coefficient (Kc), transpiration, and intrinsic water use efficiency were calculated. The information gathered from the pressure–volume curves of Cabernet Sauvignon and Tempranillo encompassed the osmotic potential at full turgor ($\gamma\pi100$), osmotic potential at turgor loss or 0 turgor ($\gamma\pi0$), water content at turgor loss (CHR0), modulus of elasticity ($\varepsilon$), and water potential at turgor loss 0 ($\gamma_H0$). The results enable a precise estimation of the water requirements for irrigation, contributing to a deeper understanding of the physiological responses of both varieties. This comprehension aids in assessing the sustainability of these vineyards amidst unexpected changes in the global mean surface temperature.

**Keywords:** temperature increase; deficit irrigation; Tempranillo; thermic stress; water stress; irrigation

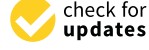



## 1. Introduction

The vine, much like other woody crops, holds significant economic importance in the Mediterranean region [1]. This reality underscores the need for a more comprehensive understanding of the vine, along with the factors influencing production and quality. The grapevine (*Vitis vinifera* L.) has historically been regarded as a dryland crop due to its re-markable adaptation to drought [2,3]. It exhibits impressive drought tolerance, enabling it to endure substantial soil water deficits [4,5].

Considering the distinctive features of the Mediterranean climate—which is marked by dry and hot summers, mild and rainy winters, and notable fluctuations in seasonal rainfall—grapevine growth is particularly crucial and sensitive to these environmental conditions. Despite these conditions, the annual rainfall frequently falls far short of the theoretical water requirements necessary for optimal plant growth that would result in a high-quality product [6,7]. The Sustainability Observatory projects a decline in the average precipitation of approximately 100 mm across various Spanish DO (Designations of Origin) in the Mediterranean area by the year 2050. This drop in production and quality will significantly impact the sector's economy [8,9]. It is anticipated that the need for irrigation

will increase (Intergovernmental Panel on Climate Change, 2014) [10], while the available water reservoirs will diminish concurrently. Consequently, a deeper comprehension of the minimal water needs for this crop has become imperative. Understanding the vineyard's hydric status and the volume of water utilized throughout the growth cycle forms the cornerstone for effective irrigation scheduling.

Various indicators within the plant characterize its water state: the movement of water within the plant (sap flow and transpiration), leaf water potential, and photosynthesis. Heat methods, such as the heat balance (SHB) method [11–15], are employed to measure sap flow in the stems. The stem water mass is determined by balancing the heat fluxes entering and exiting a stem segment. According to [16], sap measurements from the heat balance method are closely aligned (90–95%) with the daily transpiration values obtained gravimetrically. This method, being both close to reality and continuously automated, stands as the most practical choice. However, device calibration and understanding species-specific responses are essential before its generalized application.

Transpiration measurements offer a valuable means of estimating plant water consumption. Numerous studies have highlighted the substantial reduction in transpiration due to water stress [17–19]. Transpiration values display fluctuations during the vegetative cycle, exhibiting lower values in spring and during leaf fall, contrasted by higher values in summer [20–22]. Toward the end of the cycle, between maturation and harvest, reduced transpiration coincides with diminished evaporative demand due to the cessation of plant growth [23]. Varietal differences play a significant role in the impact of water deficit on transpiration. According to some sources [24–27], the variety's effect significantly influences transpiration discrepancies.

Leaf water potential, photosynthesis, and stomatal conductance [25,27] constitute common techniques for determining plant water status. Experimentally, the water potential of leaves at dawn is measured, assuming that plants are in equilibrium with the current soil potential. Diurnal fluctuations in the leaf water potential occur during situations of abundant water availability and physiological proximity to maturation. The potential, with values around $(-0.2)$ MPa at dawn, declines throughout the day in response to increasing evaporative demand, reaching values near $(-1.2)$ and $(-1.3)$ MPa [28–32]. The approximation of the water potential to its maximum value (less negative) varies based on whether the vines are irrigated or non-irrigated, as well as the stage of the growth cycle. The differences between irrigation and non-irrigation become more pronounced as maturation approaches [23,33]. Notably, the dawn potential corresponds to wet soil conditions, whereas the plant's behavior during the day mirrors a stress response [4]. In the Mediterranean region, vines often experience dry conditions and endure prolonged periods of scarce precipitation, high temperatures, and intense light exposure. Various authors have explored the impact of these stress types on photosynthesis [26,29], all agreeing that the decline in photosynthesis during water stress is attributed to stomatal closure, which prevents plant dehydration [19]. It is crucial to establish a relationship between photosynthetic activity and transpiration, yielding a parameter that signifies water use efficiency. Irrigated vines utilize more soil water without substantial effects on carbon balance, whereas stressed plants conserve water.

Summing up, it is evident that stomatal closure under water deficit conditions represents a primary limitation to $CO_2$ assimilation. Over time, the vine has managed to adapt to water deficits [34], undergoing modifications such as decreased stomatal conductance [35], rehydration ability post stress, efficient water use, and the development of negative osmotic potentials (osmotic adjustment) [36]. Osmotic adjustment plays a vital role in maintaining cell turgor as water potential decreases, thus sustaining metabolic activity, growth, and production [37]. The extent of osmotic adjustment varies among varieties [18,31,38–48]. The vine and other species have been categorized as isohydric or anisohydric, based on their stomatal responses. Isohydric plants maintain relatively constant leaf water content throughout the day, rendering them more resistant to drought. Anisohydric plants, however, do not sustain consistent foliar water potential [49,50]; their open stomata lead to

decreased leaf water potential and subsequent water stress. The classification of the most significant grape varieties for winemaking is currently a subject of considerable controversy. Many varieties that were once classified in a certain manner have exhibited opposing behaviors due to various environmental, climatic, and soil conditions.

The impact of water on the traditional components of grape quality has been a highly debated aspect of irrigation's effects in vineyards. The existing literature and studies conducted in different scenarios offer a range of diverse and occasionally contradictory outcomes [4,26,33,51,52]. It is evident, however, that negative water effects are often more attributed to a lack of understanding of the plant's genuine requirements and the optimal timing of irrigation, rather than to irrigation itself. Irrigation becomes essential during specific stages of vegetative growth and under distinct ecological circumstances that vary in each wine-producing region. Given the variability of these conditions in the Mediterranean area, further investigations are needed into plant behavior and the resultant quality effects of irrigation.

To enhance the comprehension of the mechanisms employed by red grape varieties such as Tempranillo [53,54] and Cabernet Sauvignon, which are prevalent in the Mediterranean region, these studies have explored the physiological responses to water deficits. Alongside assessing plant transpiration—a valuable tool for estimating water consumption based on soil water availability—the water potential of Tempranillo leaves was measured. This parameter offers insights into the plant's water state. Additionally, emphasis was placed on the mechanisms by which plants conserve water (stomatal closure) and changes in leaf water relations because of drought adaptation. As such, gas exchange parameters in Tempranillo were measured, along with water relations in the leaves of both varieties: osmotic adjustment capacity and shifts in cell wall elasticity that facilitate maintaining positive cell turgor as water potential diminishes. These parameters were determined using pressure–volume curves.

With these considerations in mind, the objective of this study was to assess how various deficit irrigation levels (RDI) impact water relations, transpiration, and plant metabolism in Tempranillo and Cabernet Sauvignon (*Vitis vinifera* L.) vineyards situated in the Mediterranean area.

## 2. Materials and Methods

The irrigation experiment was conducted over a period of 3 years in Tempranillo and Cabernet Sauvignon (*Vitis vinifera* L.) vineyards in the Tarragona growing area, which is in the Mediterranean region of Spain. The climate of the study area is defined as a warm Mediterranean climate and is characterized by having an average temperature of 16 °C, annual precipitation of 425 mm, annual potential evapotranspiration of 1050 mm, and it belongs to the IV region of the Winkler–Amerine scale. This area is categorized in the IV region of the Winkler–Amerine scale, indicating a warmer climate with higher cumulative temperatures that profoundly impacts vine growth and grape maturation. The moisture regime is classified as dry Mediterranean according to Thornthwaite's water table. The soil is deep, composed of calcareous and clayey material, with a basic pH (7.95) and a loamy texture. The soil classification is Calcixerept typic according to Soil Taxonomy (USDA) and Calcisol haplic according to WRB (World Reference Base for Soil Resources).

Tempranillo was chosen as it is the most important red variety in Spain. Cabernet Sauvignon was chosen because it is the most important red foreign variety in Spain. The two varieties have different adaptations to drought, and we wanted to study how they responded to different doses of irrigation in the Mediterranean area. The Cabernet Sauvignon variety is one of the most planted varieties in the world, mainly due to its rusticity, ability to adapt to different climates and soils, its regularity in harvesting, and, of course, for the quality of the grapes and wines it produces. In Spain, it represents 2.1% of the total planted vineyard area (20,000 ha). Tempranillo is one of the main varieties in the country and is considered the star of the Spanish grape varieties. It is the most cultivated, covering 220,000 ha, ahead of Grenache. The strains are vigorous, it is a productive variety

that adapts well to climates, especially cool ones, but does not tolerate extreme temperatures or water deficits well. Wines with medium/low acidity are obtained from its grapes, with discrete tannins, and all types of red and rosé wines can be made with it, from young wines to barrel-aged wines, reserve wines, and grand reserve wines.

The planting density is 3000 vines/ha, with the trellis training system in place and using the Royat pruning type. The plants grow in a trellis system consisting of three levels, the first at 60 cm from the ground and the second and third wires at 30 cm and 35 cm. The planting design is of the split-plot type, with a random distribution in blocks of 4 repetitions of Cabernet and 3 replicates of Tempranillo. In the Cabernet plot, we worked with 18 vines per irrigation treatment (6 vines/row) and in the Tempranillo plot with 24 vines/treatment (12 vines/row).

### 2.1. Irrigation Schedule

The irrigation variable was established by determining two treatments: regulated deficit irrigation (RDI) and non-irrigation (NI). The values of ET0 (reference crop evapotranspiration) were determined according to a modified Penman FAO method [54]. The crop coefficient (Kc) was used to calculate reference vine evapotranspiration. The crop coefficients (Kc) used were those proposed by [55]: budding–flowering: 0.4, drilling–pre-emerging: 0.3, germination–maturation: 0.2. The total amount of water applied for irrigation in the first year corresponded to 35% of the (ET0 × Kc). In the second year, it was decided to install the sensors in the vines irrigated with 80% of the (ET0 × Kc). With this change, we wanted to know the response of the plant's transpiration to two treatments: to a moderate stress (data from the first year with 35% of the ET0 × Kc) and to a light stress (higher doses, of 80% the second year). It was decided to apply 35% and 80% (ET0 × Kc) according to the results obtained in previous irrigation tests on the same Tempranillo variety and the same wine-growing area [56]. Plants were irrigated 3 times a week from bud burst (March) to the end of August. The total water in the three years was 196 mm, 362 mm, and 274.5 mm in the third year.

### 2.2. Transpiration Determination: Sap Flow, Leaf Water Potential, Photosynthesis, and Pressure–Volume

The parameters analyzed in this experiment were transpiration (sap flow), leaf water potential, photosynthesis, and pressure–volume curves. The study was conducted on the two chosen varieties, Tempranillo and Cabernet. More evaluations were carried out on Tempranillo than Cabernet, given that Tempranillo is the most important red variety planted in Spain and the most drought dependent. From the many data collected, only the most important results were chosen to be included in this paper. Concerning water potential, it was measured in both varieties, but only the results of Tempranillo were compared to relate them to those of the sap flow.

To determine the sap flow of the vines, sap flow meters were used following the heat balance method [13], a constant heat method developed by [13,15]. In this case, the heat source from an external annular heater is fixed and remains constant. The external heat that flows from the system is calculated by measuring temperature gradients, allowing for a direct calculation of the sap flow in the stem. Sap flow was measured using a commercial measuring system, with the heat balance method of Sakuratani. The LI-6800 Portable Photosynthesis System, a piece of infrared equipment, was used to measure the $CO_2$ assimilation rate, stomatal conductance, and transpiration.

Sap flow measurements were performed using 6 sensors. Each sensor was installed on a branch of a different vine, resulting in 3 sensors in each treatment (irrigated and non-irrigated). Before installation in the field, the device was calibrated using Tempranillo plants grown in containers. The containers were weighed every half hour for two days to determine the loss of water from the turrets by transpiration, and at the same time, transpiration data were collected with the sap flow ($kg/m^2$ of water transpired). The soil surface

of the containers was covered so that the difference in weight between measurements was due solely to the transpiration of the plant and not to the evaporation of water from the soil.

Leaf water potential was determined at pre-dawn ($\gamma_{pd}$) using a pressure chamber according to the technique of Scholander [56]. $CO_2$ assimilation rates, stomatal conductance, and transpiration were determined with LI-6400 Infrared Equipment (Li-Cor Inc., Lincoln, NE, USA).

Finally, the pressure–volume curves of the two varieties, Tempranillo and Cabernet Sauvignon, were carried out, following the methodology of [43], analyzing osmotic potential at full turgor ($\gamma\pi100$), osmotic potential at loss of turgor or turgor 0 ($\gamma\pi0$, water content at loss of turgor (CHR0), modulus of elasticity ($\varepsilon$), and water potential at loss of turgor ($\gamma_H0$).

The sap flow meter was installed from July to September in irrigated and non-irrigated Tempranillo plants, with the objective of collecting daily cycles of sap flow data at three points throughout the summer: beginning of veraison, end of veraison, and end of maturation, to showcase the evolution of the daily accumulated sap flow for the first two years of the study. Water potential (measured in MPa) was recorded at pre-dawn throughout the summer of the second and third years of the study in Tempranillo vines, both irrigated and non-irrigated. The measurements were taken on 1 and 2 July (pea size), 29 and 30 July (veraison), and 21 August (maturation). The daily cycle of transpiration (obtained using a portable photosynthesis system, LI-6800 Infrared Equipment (Li-Cor Inc., Lincoln, NE, USA)) and water potential (in MPa) of Tempranillo leaves in irrigation and non-irrigation treatments were measured on 29 July in the first year and on 21 July in the third year.

Data Analysis

Transpiration, hydric potential, and photosynthesis were evaluated through one-way analysis of variance (Factorial ANOVA); $p < 0.05$, and the Tukey's post hoc test was used with IBM SPSS Statistics software version 27.0 (SPSS version 27.0, Chicago, IL, USA). Data were replicated 3 times per treatment.

## 3. Results

The climatic data of the three years of study, shown in Table 1, is the climatic data collected at the local meteorological station located in Mas Bové, Constantí, Tarragona. According with the data from the table, the first year was a generally dry year until August. August saw the heaviest rainfall of all 3 years, but we must not lose sight of the fact that the soil was very dry given the low rainfall that accumulated since the beginning of the year. In addition, in the first fortnight of September, a relatively lower relative humidity was recorded than is normally found in the area. The second year was the one in which it rained the most: the spring showed greater precipitation and rains were spread over the 3 months of summer, which kept the soil moisture at high values when compared to what normally occurs in the area in normal years. From March to September, it rained 75% more than the previous year. As for the third year, firstly it should be noted that although it was after a wet year, it had a spring with high rainfall, but the drought in the months of July and August made it drier than the previous one.

### 3.1. Sap Flow

Before installing the sap flow meter in the field, the device was calibrated using Tempranillo plants in pots (Figure 1). A total of 75 pots were used to calibrate the sap flow. These data have been added to the manuscript, aligning with the statement: "The transpiration values measured by Sap-flow show a good correlation (73%) with the water loss obtained due to the difference in weight of the pots, considering the high number of samples used".

**Table 1.** Climatic data corresponding to the 3 years of study, with the harvest date for the two varieties.

| | T °C (max) | T °C (min) | Radiation (MJd$^{-1}$) | ET0 (mm) | Rainfall (mm) | Tempranillo Harvest Date | Cabernet Sauvignon Harvest Date |
|---|---|---|---|---|---|---|---|
| 1st Year | | | | | | 3 September | 16 September |
| Spring (March–June) | | | | | 91 | | |
| June | 26.4 | 16.1 | 23.4 | 133.0 | 3 | | |
| July | 29.5 | 16.0 | 24.8 | 10.3 | 0 | | |
| August | 29.6 | 18.3 | 20.4 | 144.9 | 66 | | |
| September | 26.0 | 16.5 | 15.1 | 97.1 | 59 | | |
| Cycle (from harvest to harvest) | | | | | | 487 | 486 |
| Yearly accumulation | | | | 982 | 395 | | |
| 2nd Year | | | | | | 1 September | 21 September |
| Spring (March–June) | | | | | 159 | | |
| June | 25.4 | 15.7 | 21.3 | 120.3 | 13 | | |
| July | 28.6 | 18.6 | 22.8 | 146.4 | 37 | | |
| August | 29.6 | 20.6 | 18.3 | 129.8 | 11 | | |
| September | 26.8 | 16.2 | 15.3 | 70.0 | 63 | | |
| Cycle (from harvest to harvest) | | | | | | 394 | 462 |
| Yearly accumulation | | | | 1002 | 372 | | |
| 3rd Year | | | | | | 14 September | 26 September |
| Spring (March–June) | | | | | 149 | | |
| June | 27.1 | 17.5 | | 142.5 | 38 | | |
| July | 29.5 | 19.8 | | 141.3 | 2 | | |
| August | 30.4 | 20.2 | | 152.4 | 12.5 | | |
| September | 27 | 17.6 | | 87.4 | 51 | | |
| Cycle (from harvest to harvest) | | | | | | 305 | 303 |
| Yearly accumulation | | | | 913 | 447 | | |

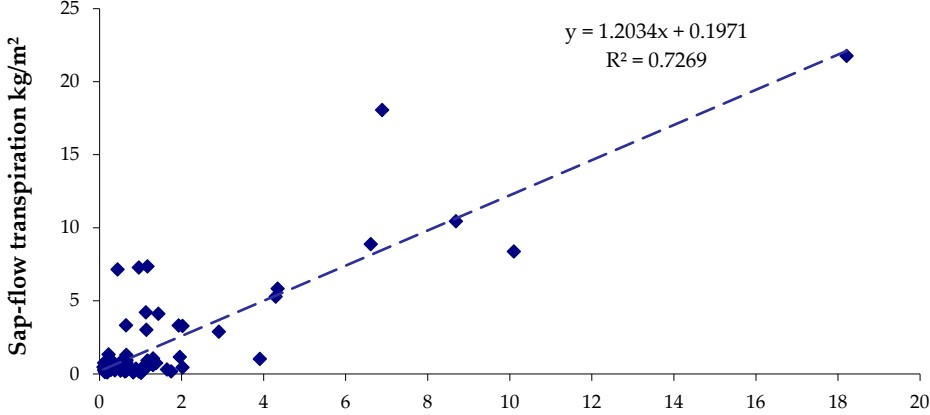

**Figure 1.** Correlation between water loss due to the weight of the containers and transpired water measured with the sap flow.

Once the instrument had been calibrated in pots, the transpiration in the field of the Tempranillo plants was measured during the two first years of the study. From the measurements of transpiration along the irrigation period, the daily cycles of transpiration and the daily accumulated sap flow values were obtained.

Figure 2 shows the daily sap flow cycles for the first year of the trial, for the two treatments. The graphs exhibit a rapid increase in the early morning hours. The maximum values are reached at 10 a.m. Transpiration, which remains constant until 2–4 p.m., depending on the time of the cycle and the treatment, decreases drastically from 4 p.m. When comparing the irrigation and non-irrigation treatments and the behaviour of the plants throughout the summer, significant differences are observed between the treatments at different times of the day and throughout the two months of study. The first year was quite

dry, and the control plants suffered from drought and reacted by decreasing transpiration. These decreases increased as the plant cycle progressed, reaching the lowest values at the end of August.

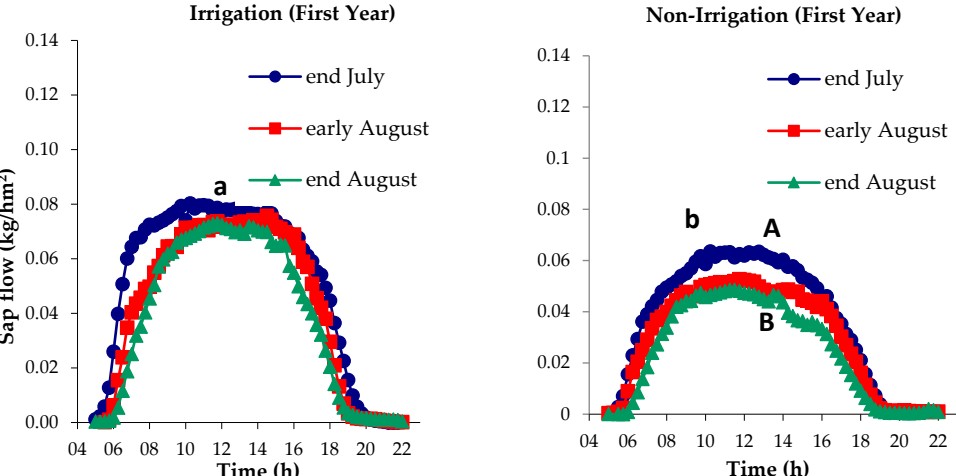

**Figure 2.** Daily cycles of sap flow corresponding to irrigation and non-irrigation, with curves representing three periods of the summer of the first year: beginning of veraison, end of veraison, and end of maturation. Solar time is indicated. Lowercase letters show statistically significant differences between irrigation and non-irrigation treatments, while capital letters show statistically significant differences between the measured points of the growing cycle: end of July, early August, and end of August.

In the second year (Figure 3), the wettest year, peaks were reached at noon, and from 12:00 p.m. to 1:00 p.m., the decrease in the sap flow began. The profile of the cycle is different from that of the previous year; the decline is not continuous, and from 2 p.m., it stabilizes, and there is even a point where a slight recovery occurs (as seen in the curve at the end of July at 4 p.m.). It appears that after the intense heat and high midday temperatures, the plant reactivates the photosynthesis process that had ceased due to stress.

At the beginning of summer, and despite not watering the control plants, the differences between the two treatments in terms of the sap flow are practically non-existent until the first half of August, with the plants transpiring at similar levels (between 0.10 and 1.20 $kg/hm^2$), and even slightly higher values were detected in non-irrigated plants. From August onward, the treatments differentiated, and until harvest, the control plants transpired less, indicating the depletion of water from the soil for the non-irrigated plants.

Given the impact of the precipitation on sap flow, particularly in the first year, which was characterized by a severe drought, the irrigated plants transpired more than the control ones. Transpiration remained without significant differences from 12 p.m. onward throughout the cycle. The control plants exhibited a decrease in transpiration over the cycle, with significant differences between July and August. During the second year, a wetter one, the plants showed higher transpiration rates than in the previous year, which was drier. The differences between the irrigated and non-irrigated treatments were less pronounced than in the drier year because the plants had access to more available water until August. The notable decrease in transpiration at the end of August in the second year (0.04 $kg/hm^2$) compared to the same period of the first year (0.06 $kg/hm^2$) can be attributed to the higher rainfall in August of the second year (66 mm compared to 11 mm in the first year, as shown in Table 1).

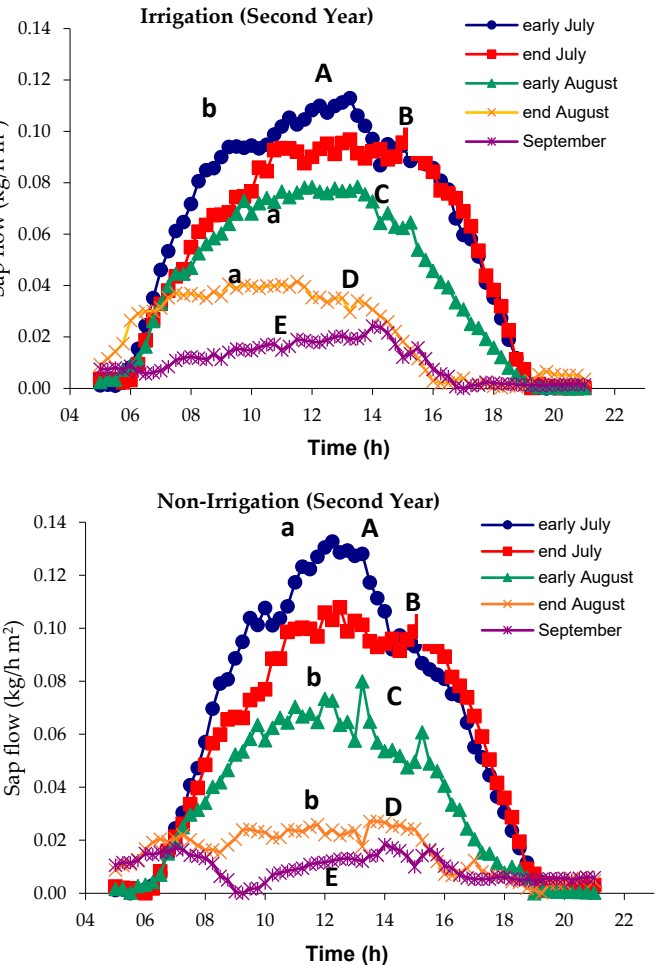

**Figure 3.** Daily cycles of sap flow corresponding to irrigation and non-irrigation treatments. The curves correspond to three distinct phases during the summer of the second year: the onset of veraison, the conclusion of veraison, and the completion of the ripening process. Lowercase letters show statistically significant differences between irrigation and non-irrigation treatments. Capital letters show statistically significant differences between measured points of the growing cycle: early July, end of July, early August, end of August, and September.

The results of the sap flow for Tempranillo measured during the summer reveal a plant response to evaporative demand and air humidity (Figure 4): the transpiration increases in both treatments as the relative humidity decreases, and it decreases when the humidity increases. On the other hand, the transpired water increases with the evaporative demand. Transpiration was highest in July and decreased until the harvest. Throughout the cycle in the first year, the non-irrigated vines transpired less than the irrigated ones, with 0.9 kg/m$^2$ and 0.6 kg/m$^2$ at the beginning of July, maintaining this average difference of 0.3 kg/m$^2$ until precipitation occurred on 3 and 29 August. During the second year, both the irrigated and non-irrigated vines increased their transpiration due to rainfall. Until the end of July, the transpired water was the same in both treatments, and in mid-August, when a significant portion of the soil water had been used up, the control plants began to show stress and their sap flow decreased to a greater extent than that of the irrigated plants, which maintained higher transpiration levels due to the water provided by the irrigation.

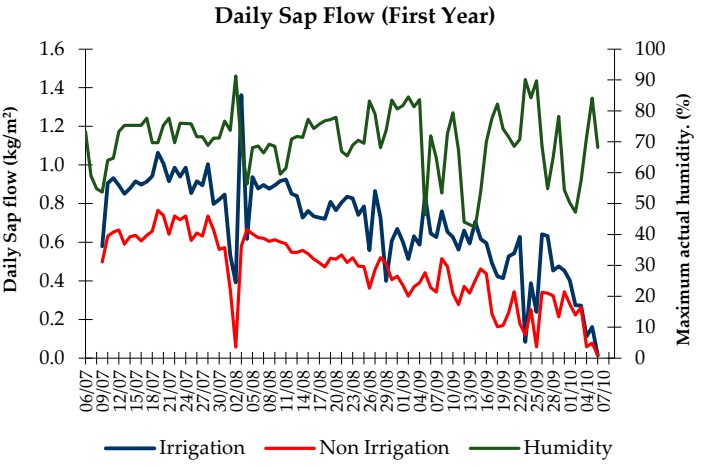

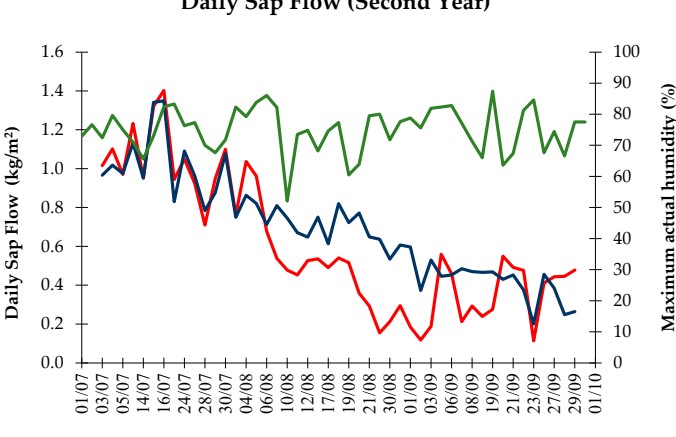

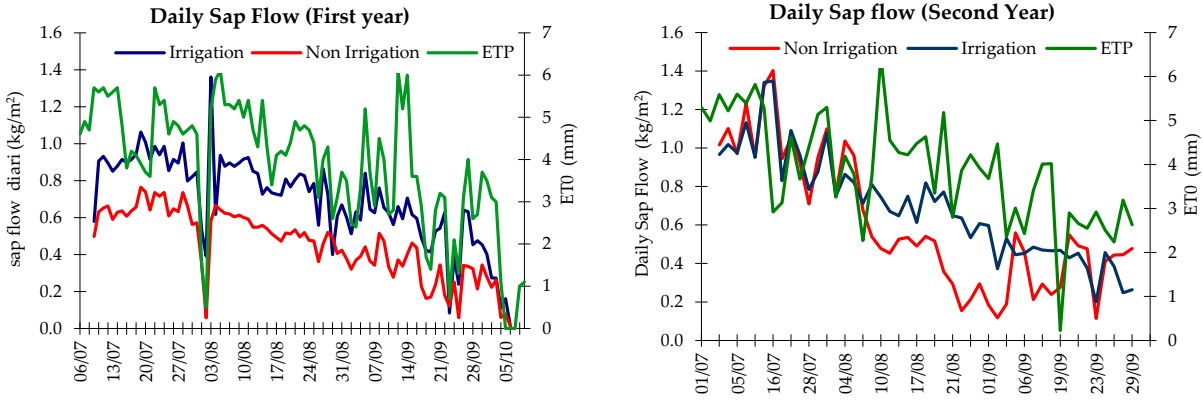

**Figure 4.** Evolution of the daily accumulated sap flow expressed per area unit, relative humidity (%), precipitation, and ET0 (mm) for the first two years of study.

Comparing the evolution of the sap flow in relation to ET0 in the second year, with a wet spring, the transpired water and evapotranspiration in both treatments were identical in the first month of measurements. During the driest periods (summer of the first year and August of the second), evapotranspiration was higher than the transpired water, as expected, but it approached the transpiration values of the irrigated plants in the days following irrigation and rain when the plants had more water available. In the case of

the control plants, after heavy rains, the transpiration values also approached those of the evapotranspiration.

In the first year, the non-irrigated Tempranillo plants transpired 46% less than the irrigated ones, as the soil was drier, and the irrigated plants retained what had been supplied through irrigation (Table 2). In the second year, the total amount of water used by the plants (T) was higher, as expected, considering that the sum of irrigation and precipitation for the period was greater. The differences between the irrigated and non-irrigated treatments in terms of the water used were only 11% (not significant). The values of the ratio of water evaporated from the soil (E) with respect to ET0 varied between 0.78 and 0.82, while the ratio of the transpired water to evapotranspiration reached the highest values in the second year, the wettest, and in irrigated plants (0.22). On the other hand, the lowest ratio was observed in the driest year (first year) and in the non-irrigated plants (0.12).

**Table 2.** Values of total transpired water in Tempranillo plants, evapotranspiration, and evaporated soil water throughout the study period. I = irrigated; NI = non-irrigated. Letters show statistically significant differences between irrigation and non-irrigation treatments. Evaporated soil water was calculated as the difference between ET0 and transpired water.

|  |  | Transpiration (T) (mm) | Evaporated Soil Water (E) (mm) | ET0 (mm) | Ratio E/ET0 | Ratio T/ET0 |
|---|---|---|---|---|---|---|
| First Year | I | 62.5 a | 290.8 b | 353.3 | 0.8 | 0.2 |
|  | NI | 42.7 b | 310.6 a | 353.3 | 0.9 | 0.1 |
| Second Year | I | 68.1 a | 247.0 a | 315.0 | 0.8 | 0.2 |
|  | NI | 61.0 b | 254.0 a | 315.0 | 0.8 | 0.2 |

When correlating the sap flow in the Tempranillo plants with the ET0 (Figure 5), it was observed that the transpiration per unit area increased linearly with the ET0, showing differential relationships for the irrigated and non-irrigated plants. In stressful situations (drier year and non-irrigated plants), less correlation is found ($R^2 = 0.75$ in the first year and in control plants) than in the conditions of water availability (irrigated plants and second year, wetter), with values of $R^2 = 0.97$ and $0.99$.

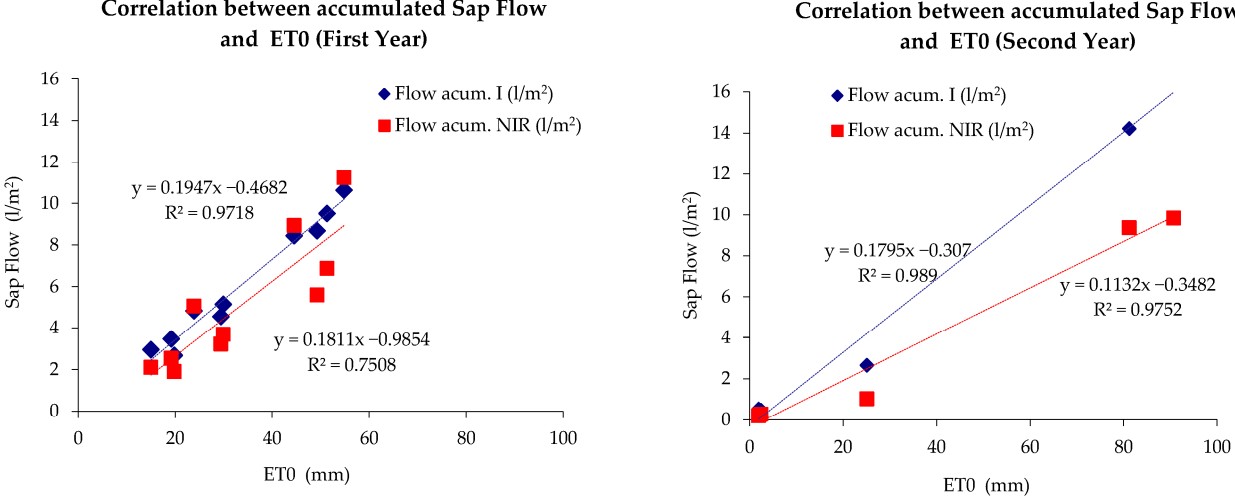

**Figure 5.** Correlation between sap flow and ET0 from the day after an irrigation (I) to the next irrigation in Tempranillo vines. Values corresponding to the first two years of study. Irrigation (I), non-irrigation (NI).

### 3.2. Estimation of Crop Coefficient Kc

Figure 6 shows the evolution of the crop coefficients, Kc, calculated for the two years of the study. The crop coefficients decrease from 0.45 and 0.60 in the first week of July in the

first and second year, respectively, to 0.35 and 0.45 at harvest time. The values are generally higher in the humid year, as expected, due to the greater availability of water.

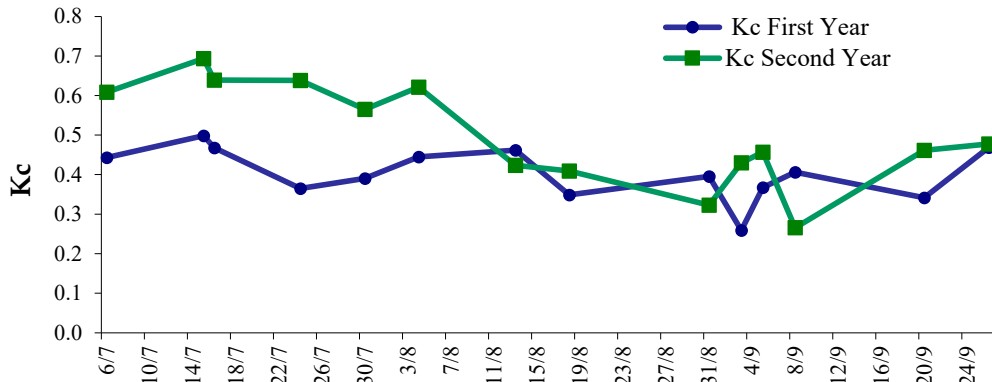

**Figure 6.** Evolution of Kc values throughout the cycle in the two years of study in Tempranillo plants.

*3.3. Physiologic Response of Tempranillo Leaves to Hydric Stress*

3.3.1. Water Potential

Figure 7 depicts the evolution of the water potential at dawn in the Tempranillo vines during the second (wet year) and third year (very dry after a wet year) of the study. During both years and in line with the environmental conditions, the water potential at dawn decreased as the dry season progressed until the end of July, with recovery beginning at the start of August. The water potentials indicated significantly different levels of plant water stress in the two treatments for each measurement day. From the second sampling of the first year and the first sampling of the second year, moderate irrigation allowed the plants to maintain more moderate potentials with less abrupt drops throughout the summer in the irrigation treatment. Stressed plants were unable to eliminate the internal water deficit during the night and showed more negative potentials. When comparing the potentials of the two years, it can be seen how the water deficit that occurred between June and the harvest of the third year enhanced the stress of the control plants, leading to higher stress levels. In general, both the irrigated and the control plants showed a water potential lower than that of the leaves in the second year.

A comparison is made between the evolution of the expanded leaf transpiration obtained with the Licor device and the evolution of the leaf water potential throughout the day for the first year and the third year (Figure 8). To understand the results, it is important to remember that the first year was characterized by a significant drought, and the third year was a dry year following a year of high rainfall (the second). The daily cycle of the transpiration and water potential in Figure 8 shows that the minimum transpiration coincided with negative potentials (after noon). In the dry year (first year), values of 0.1 L/hm$^2$ of transpiration and $-1.8$ MPa of water potential were measured for both the irrigation and non-irrigation treatments. In the wetter third year, the most negative potential values at noon ($-1.8$ MPa in non-irrigated and $-1.2$ MPa in irrigated) coincided with greater transpiration with irrigation, although the differences between the treatments were more pronounced in the water potential than in the transpiration values. A clear effect of the precipitation on the daily cycle of transpiration and water potential can be observed. The results from the plants irrigated in the driest year (first) show that the plants were more stressed than in the third year. Thus, the irrigation applied slightly improved the water status, even though the values of the third year were not reached, where the combination of the irrigation water and rainwater improved the water status of the plant.

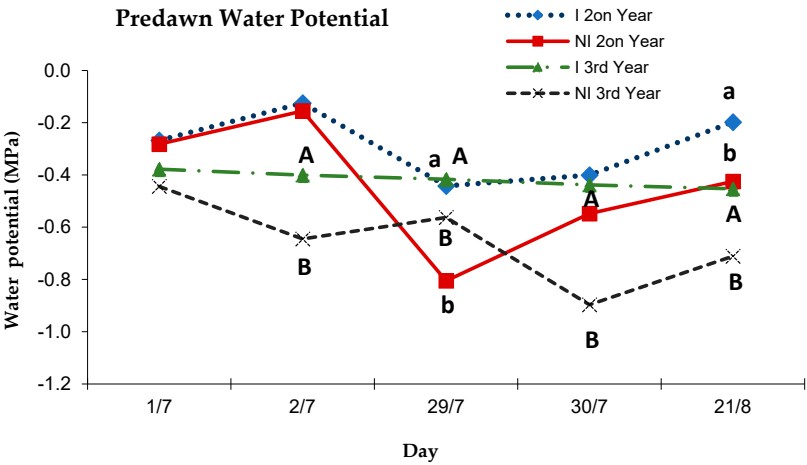

**Figure 7.** Evolution of predawn water potential (MPa) throughout the summer of the second and third years of study in Tempranillo vines. I: irrigation, NI: non-irrigation. Letters show statistically significant differences between irrigation and non-irrigation in the 2nd (lowercase letters) and 3rd years of study (capital letters).

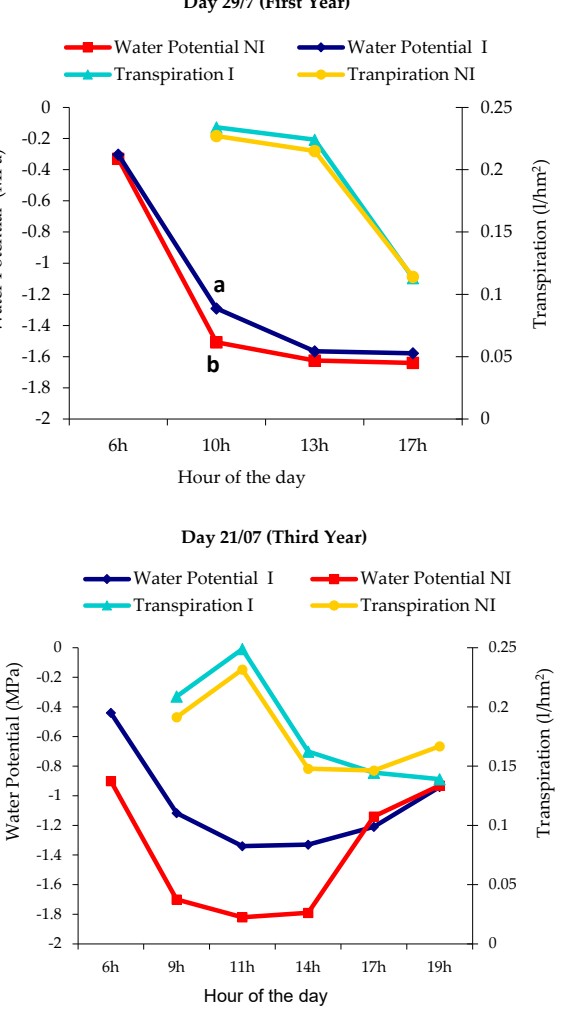

**Figure 8.** Daily cycle of transpiration (measured with Licor) and water potential (in MPa) of Tempranillo leaves in irrigation and non-irrigation treatments. Days 29/7 first year and 21/7 third year. Letters signify statistically significant differences between irrigation and non-irrigation.

### 3.3.2. Gas Exchange in Tempranillo

According to the data in Table 3, despite the good availability of water, the irrigated plants decrease stomatal conductance as the climatic conditions become more extreme throughout the day. When the irrigation and non-irrigation treatments are compared, the lower stomatal conductance values at noon and in the afternoon (1 p.m. and 5 p.m.) in the drought conditions indicate a greater control of water loss, resulting in stomatal closure. This leads to a decrease in photosynthesis, with statistically different values in the afternoon, as well as a tendency to decrease transpiration in the control plants. Tempranillo somewhat increases its intrinsic water use efficiency (expressed as the ratio between $CO_2$ exchange ($\mu mol/m^2s$) and transpiration ($\mu mol/m^2s$)) in response to the water deficit at noon and in the afternoon.

**Table 3.** Daily cycle of photosynthesis and water potential of Tempranillo. End of July of the first year. Different letters indicate statistically significant differences between irrigation and non-irrigation treatments.

| | 6 h | | 10 h | | 13 h | | 17 h | |
|---|---|---|---|---|---|---|---|---|
| | NI | I | NI | I | NI | I | NI | I |
| Intrinsic efficiency of water use (EAU) | | | $32.61 \pm 3.24$ | $35.62 \pm 4.12$ | $45.34 \pm 2.15$ b | $34.5 \pm 3.87$ a | $51.12 \pm 7.08$ b | $42.75 \pm 3.98$ b |
| Stomatal conductance ($\mu mol/m^2s$) (g) | | | $490 \pm 8.74$ | $480 \pm 9.15$ | $320 \pm 20.14$ b | $420 \pm 12.6$ a | $170 \pm 8.74$ a | $250 \pm 11.05$ b |
| Transpiration ($l/hm^2$) (E) | | | $0.227 \pm 0.13$ | $0.236 \pm 0.18$ | $0.216 \pm 0.09$ | $0.25 \pm 0.11$ | $0.11 \pm 0.04$ | $0.13 \pm 0.05$ |
| Water potential (MPa) | $-0.30 \pm 0.08$ | $-0.26 \pm 0.06$ | $-1.22 \pm 0.07$ b | $-1.37 \pm 0.06$ a | $-1.57 \pm 0.01$ a | $-1.45 \pm 0.02$ b | $-1.58 \pm 0.03$ a | $-1.43 \pm 0.02$ b |
| Assimilation of $CO_2$ (A) ($\mu mol/m^2s$) | | | $15.32 \pm 1.22$ | $16.02 \pm 1.87$ | $13.03 \pm 0.58$ | $13.24 \pm 0.45$ | $8.08 \pm 0.06$ a | $9.99 \pm 0.04$ b |

### 3.3.3. Osmotic Adjustment in Tempranillo and Cabernet Sauvignon: Pressure–Volume Curves

According to the results in Table 4, there are no differences in Tempranillo between the treatments in the $\gamma\pi100$ (osmotic potential at full turgor), in the $\gamma\pi0$ (osmotic potential at loss of turgor), or in the water content at loss of turgor (CHR0). Tempranillo seems to have no mechanisms of adaptation to drought with respect to the accumulation of solutes in the leaves, which would allow it to maintain the turgor of the cells when the water potential decreases.

Differences were observed in the modulus of elasticity ($\varepsilon$): in the drought treatment, the modulus was lower, indicating that the cell wall was more elastic. This suggests that the plant in drought conditions does not adapt very well. It can be observed that stressed plants do not show osmotic adjustment.

**Table 4.** Values obtained from the pressure–volume curves of Cabernet and Tempranillo: osmotic potential at full turgor ($\gamma\pi100$), osmotic potential at loss of turgor or 0 turgor ($\gamma\pi0$), water content at loss of turgor (CHR0), modulus of elasticity ($\varepsilon$), and water potential at loss of turgor ($\gamma_H0$). Different letters indicate statistically significant differences between irrigation and non-irrigation treatments.

| Variety | Tempranillo | | Cabernet Sauvignon | |
|---|---|---|---|---|
| Treatment | I | NI | I | NI |
| $\gamma\pi100$ (MPa) | $0.481 \pm 1.81$ | $0.462 \pm 1.34$ | $0.131 \pm 0.07$ a | $0.083 \pm 0.01$ b |
| $\gamma\pi0$ (MPa) | $0.781 \pm 0.21$ | $0.800 \pm 0.12$ | $0.156 \pm 0.05$ | $0.132 \pm 0.04$ |
| CHR0 (%) | $69 \pm 5.24$ | $66 \pm 4.06$ | $79 \pm 5.55$ | $74 \pm 6.81$ |
| $\varepsilon$ | $2.9 \pm 1.07$ a | $1.8 \pm 1.11$ b | $4.84 \pm 1.71$ | $3.21 \pm 1.87$ |
| $\gamma_H0$ (MPa) | $-1.4 \pm 0.71$ | $-0.78 \pm 0.88$ | $-1.56 \pm 0.66$ | $-1.33 \pm 0.40$ |

As for Cabernet Sauvignon, with the non-irrigation treatment (drought), turgor was lost at a lower water content than in the irrigation treatment, and therefore the plant takes longer to lose turgor in situations of water deficit. Similarly to Tempranillo, the modulus of elasticity of Cabernet Sauvignon is lower in the control, indicating that the cell wall is elastic and does not have much capacity to adapt to drought. When comparing the two varieties, Cabernet Sauvignon's values are higher than Tempranillo's, which means that although Cabernet Sauvignon's adaptation to drought is not total, it does have a somewhat stiffer wall than Tempranillo's.

Cabernet Sauvignon has uncommon values with respect to the $\gamma\pi100$ and $\gamma\pi0$ since they are higher in irrigation than in drought. High values indicate that the plant has more solutes in irrigation. In any case, it must be considered that greater water availability increases the photosynthetic capacity and foliar metabolism, and it may be that the higher solute content of these plants is due to this fact. What is clear is that Cabernet Sauvignon, like Tempranillo, does not have osmotic adjustment.

## 4. Discussion

According to the climatic data of the three years of study shown in Table 1, the first year can be classified as very dry, the second as very wet, and the third as dry but with a wet spring.

### 4.1. Sap Flow

The graphs representing the daily cycles of the sap flow (Figures 2 and 3) show differences depending on the treatment, the time of the cycle, and the year. It has already been mentioned that the first year (Figure 2) was quite dry, and the control plants suffered from drought and reacted by decreasing transpiration. On the other hand, it can be observed that the watered plants had similar sap flow values throughout the summer, except in July, when transpiration was higher in the early morning. This fact shows that these plants did not always suffer severe stress throughout the summer and that they held enough water due to the evaporative demand set by the environmental conditions. The control plants, on the other hand, showed the highest daily values of sap flow in July, decreasing progressively throughout the period. The values in August, lower than in July, coincide with the increase in the evaporative demand. In this situation, plants close stomata to avoid dehydration because of the rapid depletion of soil water caused by a combination of strongly held water and high transpiration [54,57]. At the end of August and beginning of September, when the lowest sap flow values are recorded, it coincides with the time of the cycle when the plant has stopped its vegetative growth and is in the final phase of fruit maturation [23]. As already mentioned, the profile of the cycle in the second, wetter year, was different from the first, dry year. In the second year (Figure 3), after the spring rains and before the harvest, the soil held important reserves of water at the beginning of summer. Despite not watering the control plants, the differences between the two treatments regarding the flow of sap practically did not exist until the first half of August, subsequently decreasing until the end of August when the soil reserve was exhausted. At this time, however, it was not advisable to irrigate the vineyard to preserve the quality of the grapes. On the other hand, when observing the beginning of summer, with good water reserves in the soil, the plants, both irrigated and unirrigated, transpired at similar levels (between 0.10 and 1.20 kg/hm$^2$), with even slightly higher values detected in the non-regulated plants. When the sap flows are compared with those measured by other authors in the same varieties and during the same period, it is seen that for the first year and with similar climatic conditions, the maximum values in drought are like those found by [58], in a drier Mediterranean area (Mallorca). In Mallorca, the peaks, which are also reached at 10 a.m., quickly decrease. In the present study, on the other hand, they continue until 2:00–4:00 p.m. The more pronounced stress experienced by plants in Mallorca causes them to close their stomata at noon and greatly reduce transpiration. Ref. [58] found differences of 150% in June and 100% in July and

August between the two treatments, with the irrigated plants showing higher values equal to 0.20–0.25 kg/hm$^2$, much higher than the 0.1 kg/hm$^2$ recorded in Tarragona.

The maximum daily values of irrigation flow (recorded at the beginning of the season) were around 1 kg/m$^2$ in the first year and 1.4 kg/m$^2$ in the second year (Figure 2). In drought, they were around 0.8 kg/m$^2$ and 1.4 kg/m$^2$, respectively. These maximum values are like those found by [59] in the Riesling variety in Germany, by [53] with Tempranillo, and those found by [60] with the Syrah variety in Australia. Ref. [57], with the varieties Sultana, Cabernet Sauvignon and Muscat found lower values, between 0.15 and 0.25 mm per day in measurements made in Adelaide (Australia), an area with a Mediterranean climate but with a drier summer than in Tarragona.

When evaluating the influence of the relative humidity of the air on the flow of sap (Figure 4), a close relationship is observed, since transpiration always increases in both treatments when the relative humidity decreases. Plants show the opposite behavior when humidity increases. The transpiration values of the plants responded to the changes caused by the evaporative demand. The transpired water in the two treatments was identical in the first month of measurements in year 2, coinciding with the wet period (July). During the driest periods (summer of the first year and August of the second), evapotranspiration was higher than transpired water, as expected. When there is accumulated water in the soil, practically all the ET0 is equivalent to the water transpired by the plant. The transpiration per unit area was found to increase linearly with ET0, with differential relationships for the irrigated and non-irrigated plants. For a given value, the ET0 transpiration per unit leaf area of the control plants was lower than that of the irrigated plants, because of the stomatal closure in response to the water deficit in the non-irrigated plants [58]. The more stressed the plants are, the more their transpiration decreases, and therefore, the difference with the watered plants is greater. In the first year, characterized by severe drought, the non-irrigated plants transpired 46% less than the irrigated ones, since the soil did not accumulate water and the irrigated plants used what had been supplied with the irrigation. In the second year, the total amount of water used by the plant (T) (Table 2), especially when comparing the non-irrigated treatments, was higher, as expected, considering that the sum of the irrigation and precipitation of the period was higher.

The differences between the irrigation and non-irrigation treatments in terms of the water used alone were 11% in the second year. The soil accumulated a lot of water during the winter–spring period, and therefore, the non-watered plants had enought water during the summer. It is important to highlight the reaction of the non-irrigated plants to the rainfalls, in contrast to the previous year: the increase in transpiration that occurred was greater than that of the irrigated ones. Since the control plants had plenty of water during the spring, they became used to using it without limitations and when it rained in the driest period of late July–August, the plant used all the water without regulation. The irrigations, since they had water available practically throughout the vegetative cycle (if you add the spring rain and the summer irrigations), had a higher expenditure at the end of the period (of 0.1 kg/m$^2$ daily at the beginning of August up to 0.3 kg/m$^2$ at the end of the month). Anyway, the reaction of all the plants to the rain, both the irrigated and the control ones, was not as spectacular as in the previous year. This fact corroborates that the plants, thanks to the spring and summer water, did not suffer as much stress as in year 1. Similarly, the climatic conditions of the relative humidity, temperature, and ET0 were not that high: the ET0 decreased by about 2 mm/day.

When examining the correlation of transpiration with the ET0 (Figure 5), it is observed that the transpiration per unit area increased linearly with the ET0, with differential relationships for the irrigated and non-irrigated plants. For a given value of ET0, the transpiration per unit leaf area of the control plants was lower than that of the irrigated plants, due to the stomatal closure in response to the water deficit in the non-irrigated plants [2,57]. It is apparent how the differences between the treatments increase as the evaporative demand does; the more stressed the plants are, the more their transpiration decreases, and therefore, the difference with the irrigated plants is greater.

The values of the ratio of the water evaporated from the soil (E) with respect to the ET0 (Table 2) vary between 0.78 and 0.82. This value is high, but like what other authors have found: ref. [61] found ratios of 0.77 for the Cabernet Sauvignon variety in the US. It can therefore be said that soil evaporation is the main factor in water losses. In irrigated conditions, only transpiration coincides with the ET0 when there is accumulated water in the soil; in the rest of the situations, the ET0 is higher than the transpiration. This confirms the fact already pointed out by other authors [62]: that the amount of water used by the vines is a small fraction when compared to the needs of other crops. This is like the red spruce; ref. [63] found that in irrigated conditions, the transpiration and ET0 curves coincide, and only the former is lower than the ET0 in drought situations.

The total amount of water used by the plants (Table 2) was 68.1 mm for the year 1999, which is lower than the values found by other authors. Ref. [61] speaks of 124 mm for a period of 100 days with 3-year-old Chardonnay plants. It must be considered that three-year-old plants have a lower capacity for self-regulation, and if they have water, their consumption is higher than that of adult plants. The genotypic difference must also be considered.

### 4.2. Crop Coefficient Kc

Figure 6 shows the evolution of the crop coefficients, Kc, for the vineyard, calculated for the two years of study. The values are higher in the wet year, as expected, due to the greater availability of water for the plants. The crop coefficient decreases from 0.60 and 0.45 (second and first year) in the first week of July to values close to 0.4 at the time of harvest. In relation to the calculated values represented in Figure 6, the real coefficients of the vineyard should be decreasing at the time of harvest. This reduction agrees with what [58,64] found in the Napa Valley (California), where the coefficient increases from 0.45 in the first week of July to 0.70 at the time of harvest. The crop coefficients for the vines obtained in the present study are different from those obtained by other authors [55,65]. This can be due to several factors: variation in the canopy, irrigation system, varietal and clonal differentiation, variation in the climatic conditions that determine evapotranspiration, variation in cultural practices, and soil characteristics. However, what all these authors agree on is the fact that the coefficients decrease from July to harvest.

### 4.3. Pre-Dawn Water Potential

The pre-down water potential shows the hydric state of the plant in balance with soil moisture. The water potentials indicate significantly different levels of plant water stress in the two treatments for each measurement day [66]. The evolution of the water potential at dawn throughout the summer, represented in Figure 7 for the second and third year of the study, shows that in both years, moderate irrigation allowed the plant to maintain more moderate potentials with less abrupt drops throughout the summer. Stressed plants cannot eliminate the internal water deficit during the night and show more negative potentials [67]. The initial values of the second year of study, close to ($-0.2$ MPa), corresponded to the conditions of good water availability in the soil. Conversely, the lower values in the third year indicated that the plant was stressed at dawn. These results agree with those obtained by [68], although in the study area, it was not as extreme. The sampling of the two years when the water potentials were lower than ($-1.0$ MPa) indicated a severe water deficit with drought symptoms [68]. Ref. [69] with Cabernet Sauvignon and [28,36] with Tempranillo agree in finding that during the period prior to the outbreak, the potential recovery begins at 4 p.m. (solar time), equaling the dawn value only in the case of the irrigation treatment. After the outbreak, in none of the treatments, the recovery potential is equal to the values presented at dawn. In the irrigation treatment, recovery is more important than in the non-irrigation treatment and starts two hours later. This fact highlights the greater rehydration capacity of the irrigated vines. As the season progresses, all the authors agree that the potential at dawn shows a clear decreasing trend in the non-irrigated vines, while the irrigated plants maintain more moderate potentials,

with gentle falls throughout the summer until the month of August, when an important recovery of potential begins to occur. This tells us that the plant achieves a better nighttime rehydration capacity because of irrigation, while reflecting the greater availability of water in the soil in those treatments that involve an additional contribution of water [28,36].

### 4.4. Daily Cycle of Water Potential/Transpiration

Like other authors [35], the photosynthesis rates were affected by the year. Moreover, the variation in the vine water status due to water deficit and the non-irrigation plants showed lower photosynthesis values. When comparing the evolution of the exposed leaf transpiration obtained with the Licor apparatus with the evolution of the water potential of the leaves throughout the day (Figure 8), it can be seen how both are related: in the third year of the study, the minimum transpiration values coincide with the most negative potentials. At 11 a.m., the plant has maximum transpiration and presents the lowest potentials. It quickly closes stomata, and at 2 p.m., showing the least transpiration, the water potential values are still negative. The recovery at the end of the day of both parameters in the non-irrigated plants is higher than in the irrigated ones due to better stomatal control. If the values of the treatments are compared between the years, it can be concluded that in the first year, the water applied with irrigation (35%) was not sufficient to reduce the stress, since the irrigated plants showed a similar potential to those without irrigation and more negative than that of the plants irrigated in the third year with doses of 80%. In this last year, the less negative values of the potential in the watered plants allowed the leaves to transpire more. The choice of the optimal time of irrigation has already been commented on and established based on the ET0. According to the results presented here, it would have been more accurate to determine it according to the water potential in the morning or according to the transpired water values.

### 4.5. Daily Cycle of Water Potential/Photosynthesis

Based on the data of the daily cycle of photosynthesis and the water potential observed in the first dry year (Table 3), it is evident that the irrigated plants decrease stomatal conductance as the climatic conditions become more extreme throughout the day. The stomatal conductance (gs) and water potential of the non-watered plants are lower and decrease during the day. On the other hand, in the irrigated plants, the stomatal conductance remains higher and unchanged despite the decrease in the water potential of the leaves. When comparing the irrigation and non-irrigation treatments, the lower stomatal conductance values at noon and in the afternoon (1 p.m. and 5 p.m.) in drought indicate a greater control of water losses [41,70,71]. Therefore, it can be observed that stomata close due to the extreme midday conditions to avoid leaf dehydration as a response to the water deficit [22,38,39].

As a result of stomatal closure, there is a decrease in photosynthesis, with statistically different values in the afternoon. Additionally, a tendency to decrease transpiration in the control plants was also observed by [22]. The Tempranillo variety somewhat increases its intrinsic water use efficiency, expressed as the ratio between $CO_2$ exchange ($\mu$mol/m$^2$s) and transpiration ($\mu$mol/m$^2$s), in response to the water deficit at midday and in the afternoon. Similar results have been obtained in previous studies on the same variety [36,58].

In a study of the physiological mechanisms of water use efficiency in Grenache and Syrah vineyards under drought conditions, ref. [72] also shows that the varieties considered deficit-sensitive (Syrah) show no or very little change in adaptation to stress, while those that are able to avoid drought (Grenache) have many parameters that change in response to drought in order to avoid it, such as stomatal closure in the leaves of the non-irrigated plants to maintain their hydration at levels similar to that of the irrigated plants when the water content of the soil decreases, and lower values of photosynthesis and stomatal conductance, and greater water use efficiencies.

*4.6. Pressure–Volume Curves*

Table 4 shows the values obtained from the pressure–volume curves of the two varieties, Tempranillo and Cabernet Sauvignon: osmotic potential at full turgor ($\gamma\pi100$), osmotic potential at loss of turgor or 0 turgor ($\gamma\pi0$), water content at loss of turgor (CHR0), modulus of elasticity ($\varepsilon$), and water potential at loss of turgor ($\gamma$H0). In the case of Tempranillo, it is observed that there are no differences between the treatments with respect to the $\gamma\pi100$ (osmotic potential at full turgor), $\gamma\pi0$ (osmotic potential at loss of turgor), or in the water content at loss of turgor (CHR0). This variety, therefore, seems not to have mechanisms of adaptation to drought with respect to the accumulation of solutes in the leaves; the latter situation allows for the maintenance of the turgor of the cells when the water potential decreases. Yes, differences are observed in the modulus of elasticity ($\varepsilon$): in the drought treatment, the modulus is lower; therefore, the reduced elasticity of the cell wall can impair the plant's ability to adapt effectively to drought conditions. This will make the plant in drought conditions do not adapt very well. It can be observed that stressed plants do not show osmotic adjustment. It has been seen in the literature that although changes are sometimes observed in the maintenance of turgor under conditions of stress, they are not always accompanied by a decrease in the $\gamma\pi100$ [38]. For Cabernet Sauvignon, turgor is lost in drought at a lower water content than with irrigation and therefore, the plant takes longer to lose turgor in situations of water deficit, as pointed out by [41] in the variety *Vitis vinifera* cv. Rosaki. This fact indicates that the plant tends to adapt to drought. As in the case of Tempranillo, the modulus of elasticity of Cabernet is lower in the control. The cell wall is elastic and does not have much capacity to adapt to drought. It is known that in varieties strongly adapted to water deficit, an increase in the modulus of elasticity in stressful situations indicates that the lack of water induces changes in the properties of the cell wall, making them less elastic.

When comparing, however, the values of the modulus of elasticity of the two varieties, it can be observed that those of Cabernet are higher than those of Tempranillo and this means that although the adaptation to the drought of Cabernet is not total, it has a stiffer wall than Tempranillo. A greater rigidity of the wall implies a greater sensitivity to the water deficit in the leaf, which can correspond to a tighter control of stomatal opening and therefore of water expenditure when comparing the two varieties [50]. Cabernet leaves will be able to maintain a higher turgor when they have little water, the water potentials will be less negative and therefore, this variety will have a greater ability to avoid drought than Tempranillo. Cabernet has strange values with respect to the $\gamma\pi0$ and $\gamma\pi100$ since they are higher with irrigation than in drought. In any case, it must be considered that a greater availability of water for plants increases their photosynthetic capacity and foliar metabolism, and it may be that the higher solute content of these plants is due to this fact. What is clear is that Cabernet, like Tempranillo, does not exhibit osmotic adjustment. Some authors have also found that in drought conditions, there is no osmotic adjustment in Tempranillo [36] or in other varieties: [22] in *Vitis vinifera* cv. Riesling; [25] in Syrah and Grenache, and [43] in Carignan. Other authors, however, have found the osmotic adjustment of some varieties in drought: [71] in *Vitis vinifera* Silvaner, [41,65] in *Vitis vinifera* cv. Rosaki, [73] in cv. Riesling, [36] in cv. Manto Negro, and [43] in Grenache. Chardonnay, previously classified as anisohydric [44,45], behaves as an isohydric plant according to [46]. Ref. [46] demonstrated that Syrah had an anisohydric response on the one hand and isohydric behavior on the other, when analyzing the stomatal conductance. Grenache, despite not behaving in an isohydric manner as shown by other authors, ref. [47] did find it to be a variety resistant to stress. Ref. [48], showed that the quasi-isohydric type of behavior allows Cabernet Sauvignon to maintain high photosynthesis and photorespiration. Apart from the osmotic adjustment to maintain high turgor values in the leaves of stressed vines, the presence of signals from the roots such as abscisic acid (ABA) has also been established as a mechanism of adaptation to drought (ABA), which helps control stomatal conductance and consequently photosynthesis [26]. The relationship between cell turgor and stomatal opening is known. It has been observed that in some situations, when cells lose turgor,

stomatal conductance decreases. In stressful situations, there is a decrease in conductance (Table 3). Other authors, however, in experiments that relate osmotic adjustment to the maintenance of stomatal conductance and $CO_2$ assimilation, show contradictory results on the role of leaf turgor in stomatal control: ref. [26] mentioned that stomatal response and carbon assimilation in water stress are not directly related to osmotic adjustment, developing independently. According to these authors, despite the osmotic adjustment observed in Vitis vinifera cv. Rosaki, there is a significant reduction in net photosynthesis and stomatal conductance. This decrease may be the result of chemical signals coming from the roots in dry soils.

## 5. Conclusions

In conclusion, based on these results regarding the relationships between sap flow, transpiration, and leaf water potential, it can be inferred that the entire plant sap flow system is more reliable than individual leaf gas exchange measurements, as the values from the latter significantly exceed those of the heat pulse system. The transpiration values measured by sap flow exhibit a good correlation (73%), with the water loss obtained from the difference in the weight of the pots.

The sap flow measurements have indirectly provided insights into water absorption and availability in the soil for the vines, confirming the pivotal role of the soil in irrigation decisions, timing, and dosages. For instance, the quaternary soil of our experimental plot, characteristic of many Mediterranean vineyards, contains an average water reserve in depth that facilitates water supply throughout the vegetative cycle, without reaching extreme stress conditions.

The sap flow technique is applicable for determining the amount of water used by plants, as it offers a good approximation of the response of transpiration to evaporative demand and irrigation. The gradual reduction in transpiration throughout the summer indicates that the crop coefficient (Kc) of the vine decreases from budding to harvest, with values ranging from 0.55 to 0.25.

Water potentials significantly indicate varying levels of plant water stress in the two treatments for each day of measurement and time of day. Moderate irrigation facilitates the maintenance of more moderate potentials with less abrupt drops throughout the summer in the irrigation treatment. Stressed plants are unable to alleviate the internal water deficit during the night and display more negative potentials.

The optimal timing of irrigation appears to be better established according to the γpd or transpired water determined by the sap flow, rather than not considering the ET0 values. The analysis of the pressure–volume curves lead to the conclusion that the Tempranillo variety displays poorer adaptation to drought than the Cabernet. When considering the water content at the loss of turgor between varieties, Tempranillo loses turgor at a lower water content and exhibits more solutes than Cabernet at full turgor. Consequently, the Tempranillo variety can be classified as a sensitive variety not adapted to stress, with distinct water parameters in response to drought. In contrast, Cabernet Sauvignon adapts better to drought. Neither of the two varieties exhibit osmotic adjustment against water stress.

The knowledge of the water needs of grapevine crop will be the basis for being able to act and mitigate the effects of climate change in the future. Already, in recent years, it has been observed how long-lasting heat waves, and the enduring high temperatures throughout both day and night for extended periods results in sustained evaporation from the ground, leading to significantly higher water loss than usual, despite having similar interannual precipitation. Additionally, improving soil regenerative techniques, which improves vegetative growth, reduces the risk of erosive runoff, provides a higher soil organic matter content and biological fertility, and makes irrigation management easier. Increasing resilience to drought, better water filtration, and less variable streamflow hydrology will be key factors in dealing with early water losses.

Regarding this study on the irrigation approach, grape producers can enhance sustainability by implementing agronomic measures such as optimized regulated deficit irrigation

techniques with low water volumes. Additionally, the utilization of more efficient and drought-tolerant rootstocks can significantly improve water, fertilizer, and agrochemical usage efficiencies, ultimately leading to enhanced resilience and the improved quality of grapes and wine production in semiarid regions, particularly in the context of global warming and water-limiting conditions. These practices not only mitigate the impact of water scarcity but also contribute to the overall sustainability and long-term viability of viticulture in challenging environmental conditions [74].

**Author Contributions:** Conceptualization, M.L.; methodology, M.L., M.A.M. and A.S.-O.; formal analysis, M.L. and M.A.M.; investigation, M.L.; resources, M.A.M.; writing—review and editing, M.L. and A.S.-O.; supervision, A.S.-O.; project administration, M.L. and A.S.-O. All authors have read and agreed to the published version of the manuscript.

**Funding:** This research was funded by Carburos Metálicos S.A and the National Project "Contribution of rootstocks to water and carbon balances" in National Project: structural and dynamic components and relative to production and quality grapes and wine (No. AGL2011-30408-C04-02, 2011-2014).

**Institutional Review Board Statement:** Not applicable.

**Data Availability Statement:** Data were obtained from the experimental vineyard of the public university, URV, with its permission.

**Acknowledgments:** The authors acknowledge the support of the Enology Faculty of the public University Rovira i Virgili of Tarragona, for allowing the performance of part of the experiment in its vineyard.

**Conflicts of Interest:** The authors declare no conflicts of interest.

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
