# Peer review of "Addressing Water Stress and Climate Variability in the Mediterranean: Study of Regulated Deficit Irrigation (RDI) and Non-Irrigation (NI) in Tempranillo and Cabernet Sauvignon (Vitis vinifera L.)"

_agriculture, doi:10.3390/agriculture14010129_

Round 1

Reviewer 1 Report (New Reviewer)

Comments and Suggestions for Authors

Review of agriculture-2811418: Addressing water stress and climate variability in the Mediterranean: study of regulated deficit irrigation (RDI) and non-irrigation (NI) in Tempranillo and Cabernet sauvignon (Vitis vinifera L.)

In this manuscript, the authors reported their study on the response of the Tempranillo and Cabernet Sauvignon grape varieties when subjected to deficit drip irrigation (RDI) and non-irrigation (NI), during various climatic years. I have a few main concerns as follows.

1. The sap flow meter was calibrated using water balance in the pots by covering the pots to eliminate evaporation. There were three issues that should be discussed. First, the correlation was 73%. While it might be reasonable to conclude they have similar trend, the scatter was very large, and it was quite questionable to indicate a good calibration for quantitative analysis that the study was heavily based on. Second, the magnitude from the two measured values was vastly different. The water loss of pots was only 12% of that from sap flow. Third, evaporation and transpiration should be related to each other. Covering the pots to eliminate evaporation would also affect transpiration. Therefore, the conditions in the calibration were different from the conditions that the device was used.

2. The reference and potential ET were very confusingly presented. There were four symbols: ET0, ET0, ETp and ET0p. How they were determined was not clearly explained.

3. In the Abstract, the deficit irrigation was 70% of ET0. In the Materials and Methods section, it was however 80%. The reason why 35% of ET0 was used in the second and 80% (or 70% maybe) in the second year was not convincingly justified. Different percentages in two different plots in the same three years under otherwise same conditions might be more reasonable for comparison purpose. In different years, other conditions such precipitation, temperature and other environmental conditions were so different that they might mask the difference from different levels of irrigations.

4. Language improvement is also needed as there were grammatic errors and awkward use of words throughout the manuscript.

In summary, there were some major issues with the manuscript. Therefore, I recommend major revision.

Comments on the Quality of English Language

Language improvement is needed as there were grammatic errors and awkward use of words throughout the manuscript.

Author Response

Reviewer 2 Report (New Reviewer)

Comments and Suggestions for Authors

Title

suggestion:

Addressing water stress and climate variability in the Mediterranean: in Tempranillo and Cabernet sauvignon (Vitis vinifera L.)

Introduction

The introduction is already a bit long, but 2 more paragraphs are needed describing the Tempranillo and Cabernet Sauvignon varieties. What is the characteristic of each of them? Why was it chosen? What is the main use of the fruit? Give more details...

MM

Line 136 - What is the soil classification?

Line 185 - What device was used? IRGA? It is important to describe the equipment used to determine the attributes studied.

Results

Table 1 and 2: Standardize the number of decimal places. Choose 1 or 2 decimal places. Keep it the same for everyone.

Discussion

Line 540 - But the study lasts 3 years. Why were only 2 years presented? Figures 3, 4, 5 and 6 only show 2 years.

In the discussion, I believe that there was a lack of closure to explain the social importance of the study. The authors must highlight the importance of well-done deficit irrigation management. Saving water. The preservation of natural resources resulting from water savings. Highlight the electricity savings with less irrigation on the farm. Ultimately, selling an environmentally sustainable idea.

Conclusion

Line 720 – Ok. Very good. Explore this part further.

Comments on the Quality of English Language

ok

Round 2

Reviewer 1 Report (New Reviewer)

Comments and Suggestions for Authors

Re-review of agriculture-2811418: Addressing water stress and climate variability in the Mediterranean: study of regulated deficit irrigation (RDI) and non-irrigation (NI) in Tempranillo and Cabernet sauvignon (Vitis vinifera L.)

 The authors have mostly addressed my comments, but should carefully check the manuscript to eliminate obvious errors such as:

Table 1: ET0 of July in the first year does not make sense. One digit might be missing.

The x-axis title is missing in Figure 1.

Reviewer 2 Report (New Reviewer)

Comments and Suggestions for Authors

Line 815 - "Additionally, the utilization of more efficient and 815 drought-tolerant rootstocks can significantly improve water". Cite work(s) that carried out this study.

The rest of the work was quite adequate. It's an important subject. Saving water is important.

The authors approached the topic very well!

I recommend accepting the manuscript!

Comments on the Quality of English Language

ok

This manuscript is a resubmission of an earlier submission. The following is a list of the peer review reports and author responses from that submission.

Round 1

Reviewer 1 Report

Comments and Suggestions for Authors

Dear Authors,

The paper presents important information about the grape crop grown in a deficit irrigation system. The results presented are relevant and the analysis performed is adequate. Some specific comments are shown below:

1) The introduction is too long. I suggest removing the topics/text that are not so important for the subject studied. In some sessions, repetitive information was presented, as for example in the text of lines 82-197, where a lot of information is presented about the relationship between the plant and water. Suggestion, focus on the subject presented and studied by the authors.

2) What is the hypothesis of the study? In conclusion this hypothesis must be confirmed or not.

3) The methodology section can be improved. The authors were very simplistic in describing the methodology. For example: a) which dates of planting, harvesting, were evaluated 3 years (2021, 2022?), which ones?; b) where were the meteorological data taken from?; how the management of pests and diseases was carried out....

4) Some evaluations were carried out in one cultivar and were not carried out in another, as well as in different years. Detail the existing differences.

5) In the results section, the authors present the information appropriately. Suggestion to present Anova results.

6) In the discussion section, the authors highlight the relationships obtained between plant and irrigation, however, they could deepen the approach on the climatic conditions of the three years a little more. Were conditions good or bad for the vine? What would be favorable weather conditions? stand out in the discussion.

7) Comparatively the introduction presents more information than the discussion section itself. So I suggest rethinking and emphasizing your results.

8) The conclusion is adequate.

Comments on the Quality of English Language

Minor editing of English language required.

Reviewer 2 Report

Comments and Suggestions for Authors

The manuscript is neither conceived nor written in an acceptable way. The introduction is way too long, confusing, and with a lot of punctuation problems. It must be streamlined and more focused on the topic of the paper. Moreover, the research hypothesis must be formulated more clearly.

Material and methods, on the contrary, are too short, lacking pivotal information to understand the experimental set-up. Chapter 2.1 for instance, is unreadable. Evapotranspiration is called ETc and ETP with no distinction, Kc is called cultivation coefficient (?), and the treatments are explained in a confused way with no chance to understand clearly what was done and what was the difference in the first year with respect to the second one. Which sensors were installed in the second year? How many treatments were carried out at the end? The humidity of the sun was measured?? 

Again, while reading chapter 2.2 one would expect to understand how and which method of sap flow measurements were carried out. Nothing is explained about that and also a better explanation of leaf gas exchange + leaf water potential measurements are needed (frequency, number of replicates, etc).

The presentation of the results must also be deeply improved. The plots should be presented without a title, with captions fully describing what is presented. The units of measurement should be consistent among variables, I suggest presenting everything in mm. All the speculation about cumulated fluxes and derived variables (such as evaporation in Table 2, o Kc later on) should be clearly introduced and explained in the material and method section, not in the captions. By the way the ratio E/ET looks extremely high and depends strongly on the assumption that the quantitative estimation of T with sap flow is correct, which is not sufficiently demonstrated by the correlation shown in Figure 1, which looks quite weak due to the uneven distribution of the points.

Without clearer information helping to better understand the solidity of the experimental design is impossible to further judge the discussion and the conclusion.

Comments on the Quality of English Language

Many paragraphs are poorly structured, with punctuation issues and mistakes that make the manuscript hardly readable. The overall quality of writing must be deeply improved. 

Reviewer 3 Report

Comments and Suggestions for Authors

The article “Facing water stress and climate variability in the Mediterranean: study of regulated deficit irrigation (RDI) and non-irrigation (NI) in Tempranillo and Cabernet sauvignon (Vitis vinifera 4 L.) “from Antoni Sánchez-Ortiz , Assumpta Mateos  and Miriam Lampreave evaluate evaluates the behavior of two grapes varieties irrigated with different approaches.

Overall, the article has a potential but the quality of the writing is very poor, too extensive and not precise. It is a draft of a manuscript, not sufficient to be evaluated.

In the entire document there is an extensive use of the first person. Please replace them all with the passive form and see the recent Oxford University Press discussions regarding use of the first person pronouns in scientific writing - https://blog.oup.com/2018/01/first-person-pronouns-passive-voice-scientific-writing/

Introduction

Introduction is too long, very discursive. It should be reduced maximum to one page and should contain focused information.

Material and methods

An experimental plan figure is missing, and the point of measurements are not indicated in mat e met.

The measure of the water potential is missing.

In the entire document 60cm separate number from unit measures.

Line 89  [89], The values remove the ,

Line 345:  91]: Budding remove the uppercase

Statistics is missing

Results

Results cannot start with a  table, they seems a continuation of the material and methods.

Fig. 3 2ond is not possible.

Line 424. ripening. solar time. Remove the dot.

Why the water potential  has been measured only on July 1, 2 and than only on July 29th?

The evolution of the water potential is missing, being a point measurement that does not follow the dynamic.

 Figure axis should be improved (see for example Fig. 8).

Hour. 1rst Year should be Hours (1st year)

Fig. 5 should be reconsidered, placing A) and B) with a clear legend

 Reference

Too hold, most of them are dated before 2000!

Comments on the Quality of English Language

The english should be definetly improved.